# Embryonic Ethanol but Not Cannabinoid Exposure Affects Zebrafish Cardiac Development via Agrin and Sonic Hedgehog Interaction

**DOI:** 10.3390/cells12091327

**Published:** 2023-05-06

**Authors:** Chengjin Zhang, Natalie Ezem, Shanta Mackinnon, Gregory J. Cole

**Affiliations:** 1Julius L. Chambers Biomedical/Biotechnology Research Institute, North Carolina Central University, Durham, NC 27707, USA; chengjinzhang1976@gmail.com (C.Z.);; 2Duke-NCCU Summer Scholars Program, Duke University, Durham, NC 27708, USA; 3Department of Biological and Biomedical Sciences; North Carolina Central University, Durham, NC 27707, USA

**Keywords:** cannabinoid, *shh*, *agrin*, FASD, ethanol

## Abstract

Recent studies demonstrate the adverse effects of cannabinoids on development, including via pathways shared with ethanol exposure. Our laboratory has shown that both the nervous system and cardiac development are dependent on agrin modulation of sonic hedgehog (shh) and fibroblast growth factor (Fgf) signaling pathways. As both ethanol and cannabinoids impact these signaling molecules, we examined their role on zebrafish heart development. Zebrafish embryos were exposed to a range of ethanol and/or cannabinoid receptor 1 and 2 agonist concentrations in the absence or presence of morpholino oligonucleotides that disrupt *agrin* or *shh* expression. In situ hybridization was employed to analyze cardiac marker gene expression. Exposure to cannabinoid receptor agonists disrupted midbrain–hindbrain boundary development, but had no effect on heart development, as assessed by the presence of cardiac edema or the altered expression of cardiac marker genes. In contrast, exposure to 1.5% ethanol induced cardiac edema and the altered expression of cardiac marker genes. Combined exposure to *agrin* or *shh* morpholino and 0.5% ethanol disrupted the *cmlc2* gene expression pattern, with the restoration of the normal expression following *shh* mRNA overexpression. These studies provide evidence that signaling pathways critical to heart development are sensitive to ethanol exposure**,** but not cannabinoid**s,** during early zebrafish embryogenesis.

## 1. Introduction

The effects of alcohol exposure on fetal organ development are well documented, with the incidence of fetal alcohol spectrum disorders (FASD) in humans being estimated to approach 7% of live births [1,2]. It is well established that the timing and dosage of alcohol exposure impacts the organs that are affected, with maternal alcohol consumption producing central nervous system and craniofacial defects [3,4,5] and disrupting various organogenesis pathways, including heart development [6,7].

Like ethanol, cannabinoid exposure during fetal development can induce developmental phenotypes that include FASD-like defects, which are dependent on the activation of cannabinoid receptor-1 (CB1Rs) [8,9,10]. As the use of both alcohol and marijuana is becoming more prevalent with the legalization of cannabis, studies on the effects of these two drugs on fetal organ development are of critical importance. Marijuana is now reported as the most extensively used illicit drug during pregnancy, with recent surveys indicating 2.8% of pregnant women reporting the use of marijuana during the first trimester of pregnancy [11]. In addition, 20% of pregnant women under 24 years of age have tested positive for cannabis during pregnancy [12]. Relevant to the use of both ethanol and cannabis during pregnancy, combined CB1R agonist and ethanol exposure increases the susceptibility of the neonatal brain to damage, as observed with exposure to high doses of ethanol alone [13]. In both zebrafish and mouse models, recent studies indicate that treatment with the non-selective CB1R and CB2R agonist, CP55940, produces morphological abnormalities that are remarkably similar to exposure to ethanol alone [10,14]. 

Numerous studies provide considerable evidence for ethanol exposure disrupting critical cell signaling pathways, especially extracellular growth factors which include fibroblast growth factors (Fgfs) and sonic hedgehog (Shh) [15,16,17]. In zebrafish, both Fgfs and Shh have been established as targets of ethanol exposure during embryonic development [9,18,19,20,21,22,23]. In addition, ethanol exposure in mutant mice defective in Shh signaling exacerbates the effects of ethanol, indicating a key role for Shh signaling in ethanol teratogenicity [24,25,26,27].

Since recent studies suggest that cannabinoids and ethanol may disrupt the same signaling pathways during brain development [9,10,23,28], we considered the hypothesis that both ethanol and cannabinoids may act in concert to impair the development of other organs during fetal development. With zebrafish cardiac development having been shown to be disrupted by ethanol exposure during the gastrulation and neurulation stages of embryogenesis [6,7], we decided to investigate whether these drugs interact to perturb extracellular growth factor signaling during cardiac development. As vertebrate heart development is dependent on both Fgf and Shh signaling [29], and these signaling pathways are targets of both ethanol and cannabinoids [23], it raises the possibility that heart development is sensitive to these drugs as a result of the disruption of these signaling pathways. In addition, as both CB1R (*cnr1* in zebrafish) and CB2R (*cnr2* in zebrafish) are detected in zebrafish embryos as early as approximately 10.5 h post-fertilization (hpf) [30], this suggests that the activation of these receptors may be critical to cardiac development as well. We focused our studies on the regulation of the heparan sulfate proteoglycan agrin and Shh, since our previous studies have shown that both the gene knockdown of agrin or ethanol exposure both produce cardiac edema [20,31], and Shh signaling and crosstalk with Fgfs is necessary for heart development [29]. In addition, agrin activates the ERK signaling pathway via the regulation of Fgf signaling [32] and agrin protein injection in adult mice, following myocardial infarction, and has been shown to promote heart regeneration in an ERK-dependent mechanism [33]. Thus, these data suggest agrin is a critical signaling protein in the generation of myocardiocytes and therefore could be a target of ethanol. The present studies have investigated whether both cannabinoids and ethanol disrupt agrin and Shh signaling, leading to defects in heart development. Our studies surprisingly show that cannabinoids have no effect on heart development when embryos are exposed during gastrulation and/or neurulation, possibly due to low expression levels or even the absence of CB1R and CB2R during these early stages of cardiac development. Our studies do, however, show that ethanol disrupts cardiac development as assessed by the presence of cardiac edema and altered expression patterns of cardiac marker genes, with perturbation of agrin and Shh signaling interactions representing possible molecular mechanisms for this ethanol-mediated impairment of heart development.

## 2. Materials and Methods

### 2.1. Animals

Zebrafish of the AB strain (*Danio rerio)* were obtained from the Zebrafish International Resource Center and were bred and maintained as previously described [9,21]. Fish were group-housed with <20 fish per 3 L tank on a 14:10 h (light: dark) cycle 7 days per week. All fish were fed twice daily with brine shrimp (Brine Shrimp Direct, Ogden, Utah) in the morning and dry flake fish food (TetraMin^®^, Tropical Flakes, Melle, Germany) in the evening. All protocols pertaining to the breeding and care of zebrafish were conducted according to the North Carolina Central University IACUC policy. 

### 2.2. Ethanol Treatment of Zebrafish Embryo

Zebrafish embryos in fish water, containing a 1:500 dilution of 0.1% methylene blue (to prevent fungal infection) and 0.003% 1-phenyl-2-thiourea (PTU, to inhibit pigmentation), were exposed to 0.5–1.5% ethanol from 3–24 hpf as previously described [9,21]. At least 30 embryos were treated with ethanol in each experiment. Following ethanol exposure, embryos were washed once with fresh fish water, and then transferred to fresh fish water for the remainder of the experimental time-course. All treatments were conducted with embryos with an intact chorion.

### 2.3. Cannabinoid Treatment of Zebrafish Embryos

Zebrafish embryos were exposed to 10 mg/L of the CB1R-selective agonist Arachidonoyl 2′-Chloroethylamide (ACEA) or 1.5–2.5 mg/L of the non-selective CB1R and CB2R agonist CP55940 (Cayman Chemical, Ann Arbor, MI) from 3–24 hpf, as previously described [9]. At least 30 embryos were treated with cannabinoids in each experiment. ACEA is a potent and selective CB1R agonist with similar efficacy to the endocannibinoids (eCBs). Following treatments, fish were washed with fresh fish water and then maintained until morphological analyses, as previously described [9,21]. 

### 2.4. Antisense Morpholino Injection

The properties and characterization of the antisense morpholinos (MOs) to *shh* and *agrin* (Gene Tools, Philomath, OR) used in the current studies have been previously described [21]. MOs (*shh*, *agrin* or Gene Tools control MO) were solubilized in water (0.1–1.0 mM) and injected into one-to-two cell-stage embryos. The control MO at the same concentration and volume produces no detectable zebrafish abnormalities [31]. 

For subthreshold *shh* and *agrin* MO experiments, embryos (*n* > 30) were injected with 0.3 or 1.0 pmol *shh* MO or 0.1 and 0.5 pmol of *agrin* MO, and embryos were then exposed to 0.5% ethanol from 3–24 hpf. For the *shhN183* mRNA rescue of MO/ethanol-exposed embryos, the embryos were co-injected with 0.3 pmol *shh* MO or 0.1 pmol *agrin* MO, and 25 pg *shhN183* mRNA. *ShhN183* mRNA was synthesized according to previously published methods [21]. PCR primers for *shhN183* were CGGAATTCATGCGGCTTTTGACGAGAGTG and GCTCTAGATCAGCAATGAATGTGGGCTTTGG. The PCR product was subcloned into pCS2 vector and confirmed by sequencing. Capped mRNA was synthesized with Ambion cap mRNA kit.

### 2.5. Midbrain–Hindbrain Boundary Analysis

Midbrain–hindbrain boundary (MHB) development and malformation were scored at 1 dpf, as previously described [34]. Malformation of the MHB was assessed visually based on the absence of the defined border between the midbrain and hindbrain. The presence of a normally formed MHB was defined as the presence of 3 or 4 ridges (tectal and cerebellar boundaries) perpendicular to the anterior–posterior axis of the CNS at the midbrain–hindbrain junction. Absence of this defined border was scored as representing the disruption of MHB development.

### 2.6. Whole-Mount In Situ Hybridization

Whole-mount in situ hybridization was carried out as previously described [31,35]. Digoxygenin-labeled riboprobe to *cardiac myosin light chain 2* (*cmlc2)*, *nkx2.5* or *(ventricular myosin heavy chain (vmhc)* was transcribed as previously described [21]. cDNA constructs for *cmlc2*, *nkx2.5* and *vmhc* were a generous gift from the James A. Marr lab at IUPUI School of Science. All embryos (control and treatments) were developed for riboprobe staining for an identical amount of time, and were analyzed to score embryos for altered or cardiac marker mRNA staining. A second laboratory member was blinded to the treatments and their scoring of staining levels was the same as the unblinded laboratory member. The statistical significance of treatment differences was determined using an unpaired *T*-test.

For visualization of *cnr1* and *cnr2* mRNA expression, PCR primers were generated for *cnr1*, GCTGAAGACGGCAGTCTGC and TAATACGACTCACTATAGGGAGACAGCGTCTTGGCCAGACGG, and *cnr2* (CGGAATTCATGGAGAACAAACTGGAACAAG and CCGCTCGAGTTTTGCTGCCTGTCTGCAC), and used to prepare digoxygenin-labeled riboprobe, as previously described [21].

## 3. Results

### 3.1. Effects of Ethanol and Cannabinoids on Zebrafish Embryo Development

Previous studies from our laboratory and others have shown that endocannabinoid agonists such as ACEA and CP55940 perturb CNS development, in particular producing a small eye phenotype [9,10,23]. As a first step in assessing whether cannabinoids and ethanol may interact to impact heart development, we analyzed whole body embryos for any obvious morphological phenotypes, as it is well documented that ethanol produces small eyes, microcephaly, and even heart edema. We used a chronic ethanol exposure of 3–24 hpf with 1.5% ethanol to ensure the complete penetrance of the drug exposures, with all embryos exhibiting gross dysmorphologies. We observed complete penetrance with this ethanol exposure, with 100% of embryos exhibiting dysmorphologies. As shown in Figure 1, 1.5% ethanol exposure produced microcephaly, reduced eye size, and heart edema (arrow in Figure 1G). However, neither a high dose of ACEA (based on our previous studies; Boa-Amponsem et al., 2019) nor a range of CP55940 doses produced heart edema in embryos (Figure 1H–J). To demonstrate that the concentrations of cannabinoid agonists used were in fact capable of producing dysmorphologies, we examined MHB development at 1 dpf. As shown in Figure 1B,C,E, ethanol, the cannabinoid agonist ACEA, and the higher dose of CP55940, disrupted MHB formation. A total of 30/30 embryos exhibited disrupted MHB formation with ethanol or 2.5 mg/L CP55940 exposure, 26/30 embryos exhibited disrupted MHB with ACEA exposure, and only 2/30 embryos displayed a disrupted MHC with 1.25 mg/L CP55940 exposure. Thus, these experiments show that while ethanol exposure from 3–24 hpf produces marked heart edema, cannabinoid exposure does not result in heart edema.

While exposure to cannabinoid receptor agonists does not induce gross dysmorphologies such as cardiac edema, it is still possible that these agonists perturb the normal cellular differentiation program in cardiomyocytes. We therefore examined the expression of cardiac marker genes following exposure to either ethanol or cannabinoid receptor agonists, first analyzing the expression of *cardiac myosin light chain 2* (*cmlc2*), which is expressed throughout the early myocardium in both the atrium and ventricle [36]. In a dorsal view of 1 dpf embryos, *cmlc2* mRNA expression is seen in the fused myocardium as a linear heart tube that has begun to be formed (Figure 2A), with 1.5% ethanol exposure from 3–24 hpf delaying the formation of the cardiac cone and fused myocardium, and with an unfused myocardium being observed (Figure 2B). The main effect of the ethanol treatment is a change in the pattern of *cmlc2* gene expression. In contrast, both ACEA and CP55940 exposure have no effect on the development of the early heart based on the pattern of *cmlc2* gene expression (Figure 2C–E). In 1 dpf embryos, when viewed laterally, the linear heart tube is apparent and ethanol alters the *cmlc2* mRNA expression pattern, indicative of a lack of looping in the unfused myocardium (Figure 2F,G). Again, no effect is observed following cannabinoid agonist exposure in either 1 dpf or 3 dpf hearts (Figure 2H–J,H’–J’). However, in the 3 dpf embryo, an alteration in *cmlc2* expression persists with ethanol exposure, with a lack of looping in the linear heart tube (Figure 2G’). 

As the *cmlc2* mRNA expression pattern was altered in early heart development following ethanol exposure, but not cannabinoid exposure, we decided to examine additional cardiac markers to confirm the lack of effect of cannabinoid receptor agonists on early heart development. We selected *nkx2.5*, a transcription actor that is expressed throughout the developing myocardium, and *vmhc (ventricular myosin heavy chain)*, whose expression is restricted to the ventricle. Consistent with our results when examining *cmlc2*, we only observed an alteration in the *nkx2.5* gene expression pattern in 1 dpf embryos with ethanol exposure (Figure 3B). Similarly, we only observed a change in the *vmhc* expression pattern in 1 dpf embryos with ethanol exposure (Figure 3G).

Since our cannabinoid and ethanol exposure was from 3–24 hpf, we considered the possibility that the cannabinoid receptor agonists had a decrease in potency with long-term incubation, which was especially relevant based on previous studies suggesting that the cannabinoid receptor expression in zebrafish embryos was first detected at approximately 10.5 hpf [30]. To address this possibility, we carried out a series of experiments examining *cmlc2* mRNA expression with embryos exposed to ethanol or cannabinoid receptor agonists from 10–24 hpf. We also increased the concentration of ACEA and CP55940 used to treat embryos to ensure that we would observe any effects of these agonists on cardiac development. As shown in Figure 4, again, only ethanol exposure resulted in alterations in the *cmlc2* gene expression pattern in both 1 dpf and 3 dpf embryos (Figure 4B,H,H’). Thus, these studies suggest that cannabinoid receptor activation is not necessary for early cardiac development in zebrafish.

### 3.2. Analysis of Cannabinoid Receptor Gene Expression in Zebrafish Embryos Using In Situ Hybridization

Since previous studies indicated that the cannabinoid receptor expression in zebrafish embryos was first detected by qRT-PCR at approximately 10.5 hpf [30], we examined *cnr1* and *cnr2* gene expressions in zebrafish embryos to ascertain if these receptors are expressed in the developing heart. In both 1 dpf and 3 dpf embryos, we detected the marked expression of both receptor mRNAs in the head region (Figure 5), especially in brain and eye for *cnr1* (Figure 5A–C). For both receptor genes, we did not detect expression in heart, even with increased staining times and a higher magnification analysis of the heart region in 3 dpf embryos (Figure 5C,C’,F,F’). Thus, the absence of effects of cannabinoid agonists on early heart development may be due to either an absence of expression or low levels of expression not readily detectable by in situ hybridization.

### 3.3. Does Ethanol Impact Cardiac Development via Impaired Extracellular Matrix Signaling?

As previous studies have documented the role of extracellular matrix signaling as a target of ethanol teratogenicity, involving Fgf and Shh as well as molecules such as agrin that modulate Fgf and Shh signaling [20,21], and cardiac development requires an intricate crosstalk between these extracellular signaling pathways, we postulated that the ethanol disruption of cardiac development would also be linked to an impairment of these extracellular signaling pathways. Using a morpholino (MO) approach to partially knockdown the target gene expression, our laboratory has previously documented the importance of agrin, Fgf and Shh signaling in ethanol effects on CNS development in zebrafish [20,21]. Using a high dose of either *agrin* (0.5 pmol) or *shh* (1.0 pmol) MO, we observed a similar high disruption of the cmlc2 gene expression in 1 dpf embryos as was observed with 1.5% ethanol exposure, with 55/60 of embryos exhibiting an altered *cmlc2* expression with *agrin* MO, and with 49/60 embryos affected with *shh* MO (Figure 6C,D and Figure 7). While neither 0.5% ethanol exposure nor low-dose *agrin* or *shh* MO alters the *cmlc2* expression pattern in 1 dpf embryos (Figure 6B,E,G), combining 0.5% ethanol with 0.1 pmol *agrin* MO or 0.3 pmol *shh* MO produced a *cmlc2* expression pattern similar to 1.5% ethanol treatment (Figure 6F,H), with 41/60 of embryos displaying an altered *cmlc2* expression with combined *agrin* MO and ethanol (Figure 6F) and 37/60 embryos displaying an altered *cmlc2* expression with combined *shh* MO and ethanol (Figure 6H). In addition, the overexpression of *shhN183* mRNA rescued both the combined ethanol and *agrin* or *shh* MO effects, with only 11/60 embryos displaying an altered *cmlc2* gene expression pattern with the rescue of *agrin* MO + ethanol embryos (Figure 6J), and only 10/60 embryos following rescue with *shh* MO + ethanol-treated embryos (Figure 6L). Extending this gene expression analysis to 3 dpf embryos, we observed a similar effect of combined low-dose ethanol and *agrin* or *shh* MO on *cmlc2* gene expression, with 31/60 *agrin* MO + ethanol embryos exhibiting an altered *cmlc2* expression pattern, and 26/60 *shh* MO + ethanol embryos exhibiting an altered *cmlc2* gene expression pattern (Figure 8F,F’,H,H’ and Figure 9). Likewise, we observed a rescue of the combined ethanol and *agrin* or *shh* MO effects by *shhN183* mRNA overexpression, with only 9/60 embryos having an altered *cmlc2* pattern with the rescue of *agrin* MO + ethanol embryos, and 7/60 embryos having an altered *cmlc2* pattern with the rescue of *shh* MO + ethanol embryos (Figure 8J,J’,L,L’). 

## 4. Discussion

The basis for the current studies was to ascertain whether cardiac development, like eye and brain development, is sensitive to both cannabinoid and ethanol exposure as a result of perturbed signaling by extracellular signaling proteins that include Shh and the heparan sulfate proteoglycan agrin. A plethora of studies have demonstrated that prenatal exposure to alcohol [3,25] or cannabinoids [14,37] causes developmental deformities, although the mechanisms underlying their effects are unclear [37]. However, as recent studies have suggested that cannabinoids act as inhibitors of Smoothened and Shh signaling [28], and that ethanol and cannabinoids decrease Shh or Fgf signaling to produce FASD phenotypes [9,10,23], then the dependence of cardiac development on Fgf and Shh signaling would suggest that cannabinoid receptor activation would also produce FASD cardiac phenotypes.

Our studies provide the unexpected observation that unlike CNS development, cardiac development in zebrafish is not impaired as a result of the activation of cannabinoid receptors. While several explanations could account for our observed lack of effects of cannabinoids on heart development, such as cannabinoid receptor activation not contributing to early cardiac development, our analysis of *cnr1* and *cnr2* gene expressions suggests that cannabinoid receptors are not expressed at significant levels in early zebrafish heart. The use of chronic cannabinoid receptor agonist treatments, as well as concentrations that disrupt CNS development, strongly suggests that the most likely explanation for the absence of effects of the agonists on heart development is that *cnr1* and/or *cnr2* are not expressed in heart during early development. Although it is unclear if other vertebrate species will also display low or no cannabinoid receptor expressions in early heart, at least in zebrafish, it appears that endogenous cannabinoid signaling does not play a significant role in early heart development. It will be of interest in future studies to ascertain when *cnr1* and/or *cnr2* genes are expressed in heart muscle, and whether ethanol perturbs this cannabinoid signaling. This is noteworthy since studies suggest that CB1R (*cnr1* in zebrafish) activation plays an important role in diabetic cardiomyopathy via facilitating ERK activation [38].

While our studies show that ethanol exposure alters cardiac development, consistent with previous studies [6,7], our studies do provide new information relating to the potential molecular targets of ethanol in cardiac development. Our combined low-dose ethanol and low-dose MO experimental paradigm demonstrates that both agrin and Shh functions are required for normal early heart development, and that ethanol disrupts the signaling of these extracellular proteins. As previous studies have shown that eye, forebrain and hindbrain development require a crosstalk between these signaling pathways, which is disrupted by ethanol exposure [9,15,16,17,18,19,20,21,22,23], the results from the analysis of early cardiac development suggest that multiple organs that are negatively impacted by fetal ethanol exposure use shared cell signaling pathways to guide organ development. Limb development also requires a cascade of signal pathway activation involving Fgfs and Shh [39,40], which is disrupted by ethanol [26,41], again indicating that a common mode of teratogenicity is used by ethanol to disrupt the normal development of multiple organ systems in FASD. While we did not investigate the role of Fgf signaling in the present study, our previous studies have shown that Fgf and Shh interact to regulate ethanol effects in zebrafish brain [21,23], and we have observed that *fgf8* MO will alter the *cmlc2* expression in an identical manner as observed with *agrin* or *shh* MO.

The rationale for examining the role of agrin in ethanol’s effects on early heart development is that agrin is a major basement membrane heparan sulfate proteoglycan [42,43] that binds to numerous extracellular matrix signaling molecules, including Fgfs, Shh and other heparin-binding growth factors [44,45]. *Agrin* knockdown in zebrafish perturbs organ development that is dependent on Fgf and/or Shh signaling [21,31,35], and marked cardiac edema is observed with *agrin* knockdown in zebrafish [31]. Agrin activates the ERK signaling pathway via the regulation of Fgf signaling [32] and agrin protein injection in adult mice, following myocardial infarction, has been shown to promote heart regeneration in an ERK-dependent mechanism [33]. Thus, these data suggest that agrin is a critical signaling protein in the generation of myocardiocytes. Our present studies show that *agrin* MO treatment disrupts the normal expression pattern of *cmlc2*, resulting in an unfused myocardium, thus suggesting a delay in early heart development. Subthreshold doses of *agrin* MO, that do not alter *cmlc2* expression, are capable of disrupting *cmlc2* when combined with a subthreshold dose of 0.5% ethanol. Thus, our studies indicate that multiple extracellular matrix signaling proteins may be critical targets of ethanol teratogenicity, suggesting that polymorphisms and/or mutations in multiple candidate genes could place a fetus at risk for FASD if exposed to ethanol.

## 5. Conclusions

In summary, the present studies set out to determine if cannabinoids would interact with ethanol to disrupt normal cardiac development, as has been observed in the CNS. In addition, as both cannabinoid and ethanol perturb Shh signaling, which is also regulated via interactions with other signaling proteins such as agrin and Fgfs, a goal of these studies was to determine if the ethanol-mediated disruption of cardiac development was due to impaired extracellular matrix signaling. While our studies indicate that, at least in zebrafish, cannabinoid receptor activation is not a regulatory mechanism in cardiac development, our data also provide new information underlying the molecular pathways perturbed by ethanol that result in heart defects. Our studies suggest that shared mechanisms underlie ethanol’s teratogenicity, of which leads to the disruption of the normal development of multiple organ systems, including cardiac development. Future studies are warranted to determine when cannabinoid receptors may become important in cardiac function, especially since adult disorders, such as diabetic cardiomyopathy, are dependent on cannabinoid receptor activation in an ERK-dependent mechanism [38], and agrin is a biomarker for diabetic nephropathy [46].

## Figures and Tables

**Figure 1 cells-12-01327-f001:**
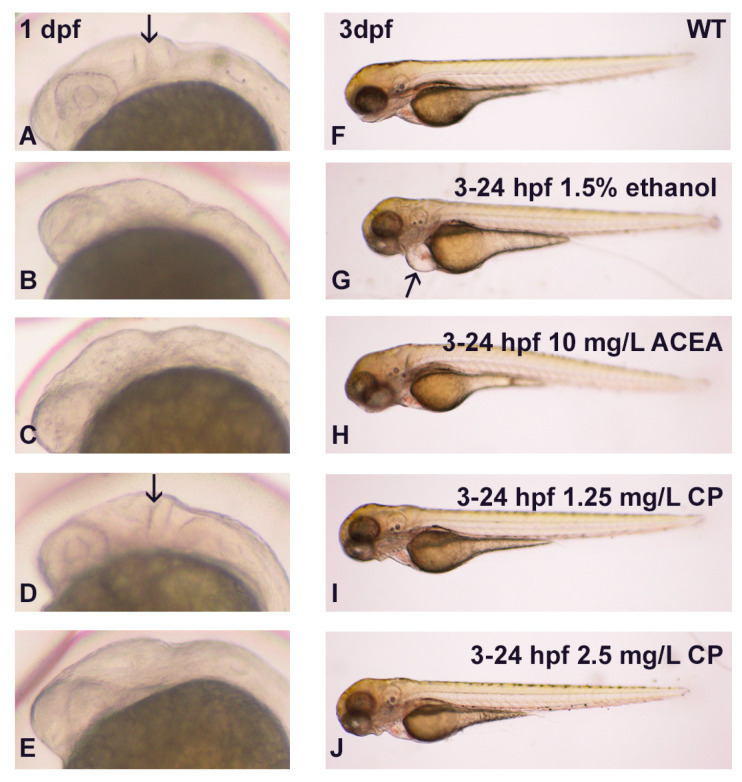
The effects of ethanol and cannabinoids on MHB and cardiac development. (**A**–**E**), MHB development was disrupted with 3–24 hpf 1.5% ethanol or 10 mg/L ACEA or 2.5 mg/L CP exposure (**B**,**C**,**E**), but not with 1.25 mg/L CP exposure ((**D**), arrow). (**F**–**J**), Obvious pericardial edema was only observed with exposure to 1.5% ethanol ((**G**), arrow), with no edema observed with any of the cannabinoid doses.

**Figure 2 cells-12-01327-f002:**
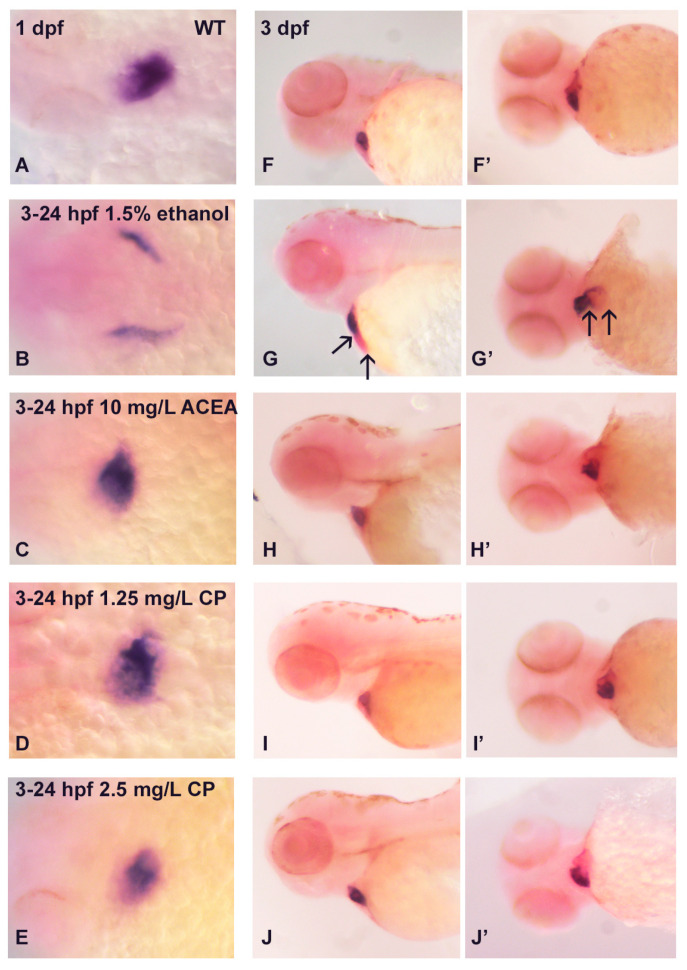
Effects of ethanol and cannabinoid exposure on the expression of *cmlc2* in embryonic zebrafish heart. Compared to the *cmlc2* expression in untreated embryos ((**A**), dorsal view, (**F**,**F’**) lateral view), only exposure to 1.5% ethanol disrupts the normal *cmlc2* cardiac expression pattern ((**B**,**G**,**G’**), arrows) in 1 dpf and 3 dpf embryos. No effects of cannabinoid exposure on *cmlc2* expression were observed at any concentrations of these agonists (**C**–**E**,**H**–**J**,**H’**–**J’**). Each experiment was conducted in triplicate with an *n* = 20, with 60/60 embryos showing a normal *cmlc2* expression pattern with no treatment or cannabinoid agonist exposure, and 0/60 embryos showing a normal cmlc2 expression pattern with ethanol exposure.

**Figure 3 cells-12-01327-f003:**
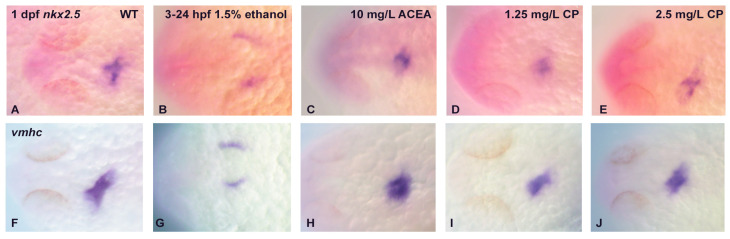
The effects of ethanol and cannabinoid exposure on the expression of other cardiac markers in embryonic zebrafish heart. The expressions of *nkx2.5* (**A**–**E**) and *vmhc* (**F**–**J**) were examined in 1 dpf heart following ethanol or cannabinoid exposure from 3–24 hpf. Compared to the marker expression in untreated embryos (**A**,**F**), only exposure to 1.5% ethanol disrupts the normal cardiac marker expression pattern (**B**,**G**), with no effects of cannabinoid exposure on cardiac marker expression observed at any concentrations of these agonists (**C**–**E**,**H**–**J**). Each experiment was conducted in triplicate with an *n* = 20, with 60/60 embryos showing a normal *nkx2.5* or *vmhc* expression pattern with no treatment or cannabinoid agonist exposure, and 0/60 embryos showing a normal marker gene expression pattern with ethanol exposure.

**Figure 4 cells-12-01327-f004:**
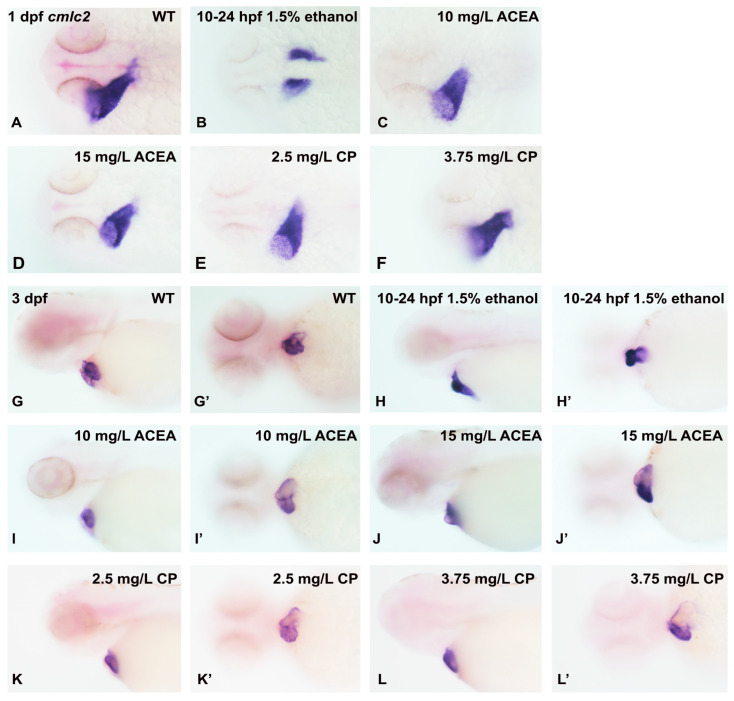
The effects of ethanol and cannabinoid exposure on the expression of the *cmlc2* cardiac marker in embryonic zebrafish heart following drug exposure from 10–24 hpf. The expression of *cmlc2* was examined in 1 dpf (**A**–**F**) heart (dorsal view) and 3 dpf (**G**–**L’**) heart (ventral and lateral view) following ethanol or cannabinoid exposure from 10–24 hpf. Compared to the marker expression in untreated embryos (**A**,**G**,**G’**), only exposure to 1.5% ethanol disrupts the normal *cmlc2* expression pattern (**B**,**H**,**H’**), with no effects of cannabinoid exposure on expression observed at any concentrations of these agonists (**C**–**F**,**I**–**L’**). Each experiment was conducted in triplicate with an *n* = 20, with 60/60 embryos showing a normal *cmlc2* expression pattern with no treatment or cannabinoid agonist exposure, and 0/60 embryos showing a normal *cmlc2* expression pattern with ethanol exposure.

**Figure 5 cells-12-01327-f005:**
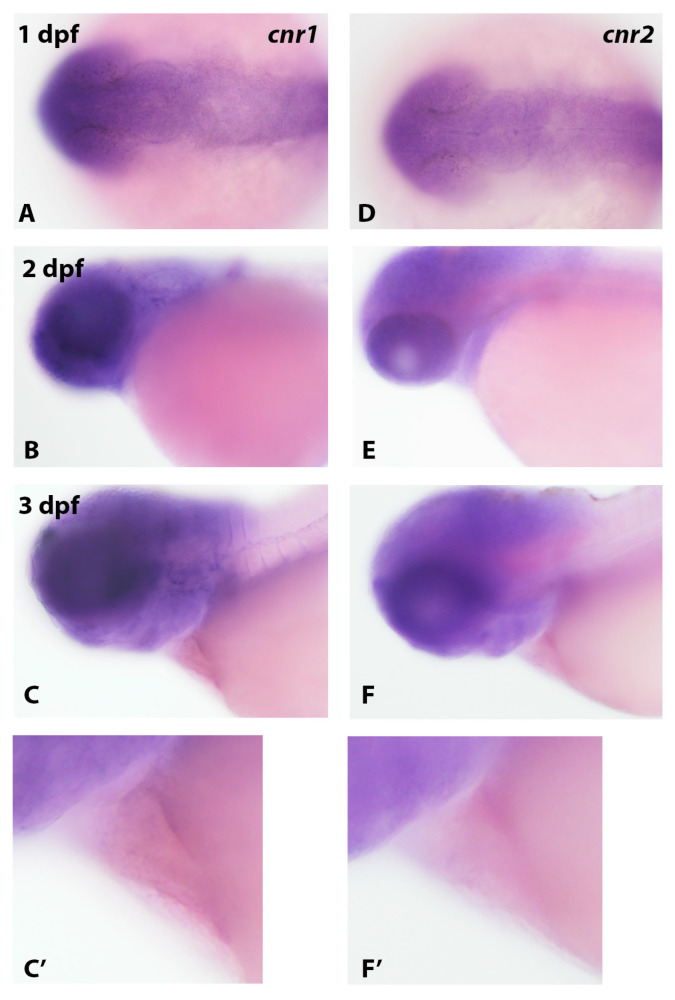
Cannabinoid receptor expression in early zebrafish development. The expression pattern of *cnr1* (**A**–**C’**) and *cnr2* (**D**–**F’**) in 1 dpf, 2 dpf and 3 dpf embryos was analyzed by in situ hybridization. A robust expression was observed in the head region of embryos (**A**–**F**) with an apparent lack of expression in early heart (**C**,**C’**,**F**,**F’**).

**Figure 6 cells-12-01327-f006:**
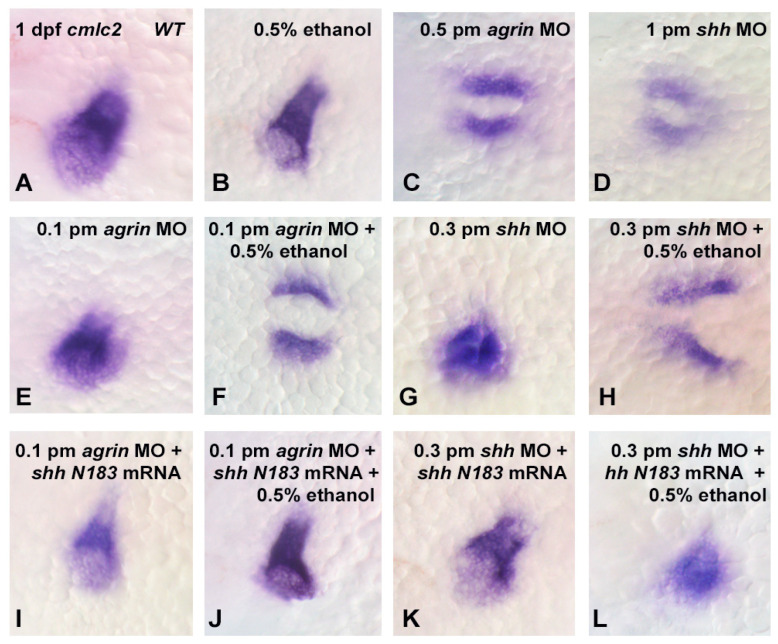
Extracellular matrix signaling involving agrin and Shh is disrupted by ethanol exposure in early cardiac development. *Cmlc2* mRNA expression was analyzed in dorsal views of 1 dpf embryos. (**A**), wild-type; (**B**), 0.5% ethanol exposure from 3–24 hpf; (**C**), 0.5 pmol *agrin* MO injection of embryos; (**D**), 1 pmol *shh* MO injection of embryos; (**E**), sub-threshold 0.1 pmol *agrin* MO injection; (**F**), 0.1 pmol *agrin* MO combined with 0.5% ethanol exposure; (**G**), sub-threshold 0.3 pmol *shh* MO injection; (**H**), 0.3 pmol *shh* MO combined with 0.5% ethanol; (**I**), 0.1 pmol *agrin* MO and 25 pg *shhN183* mRNA injection of embryos; (**J**), 25 pg shhN183 mRNA injection rescues the *cmlc2* expression pattern in embryos treated with 0.1 pmol *agrin* MO and 0.5% ethanol; (**K**), 0.3 pmol *shh* MO and 25 pg *shhN183* mRNA injection of embryos; (**L**), 25 pg *shhN183* mRNA injection rescues *cmlc2* expression pattern in embryos treated with 0.3 pmol *shh* MO and 0.5% ethanol.

**Figure 7 cells-12-01327-f007:**
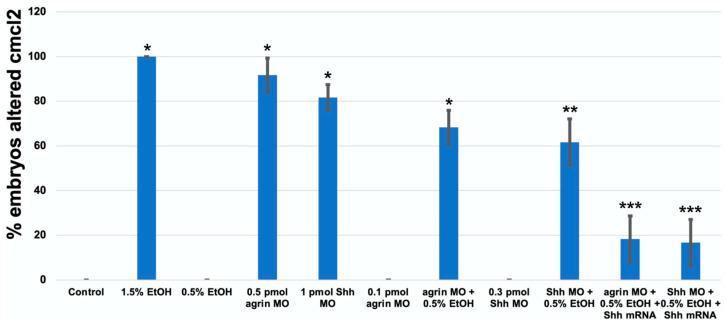
*Cmcl2* mRNA decreases, resulting from MO and ethanol exposures in 1 dpf embryos. * Significantly different from control *p* < 0.01; ** *p* < 0.025; *** Significantly different from MO + EtOH, *p* < 0.025.

**Figure 8 cells-12-01327-f008:**
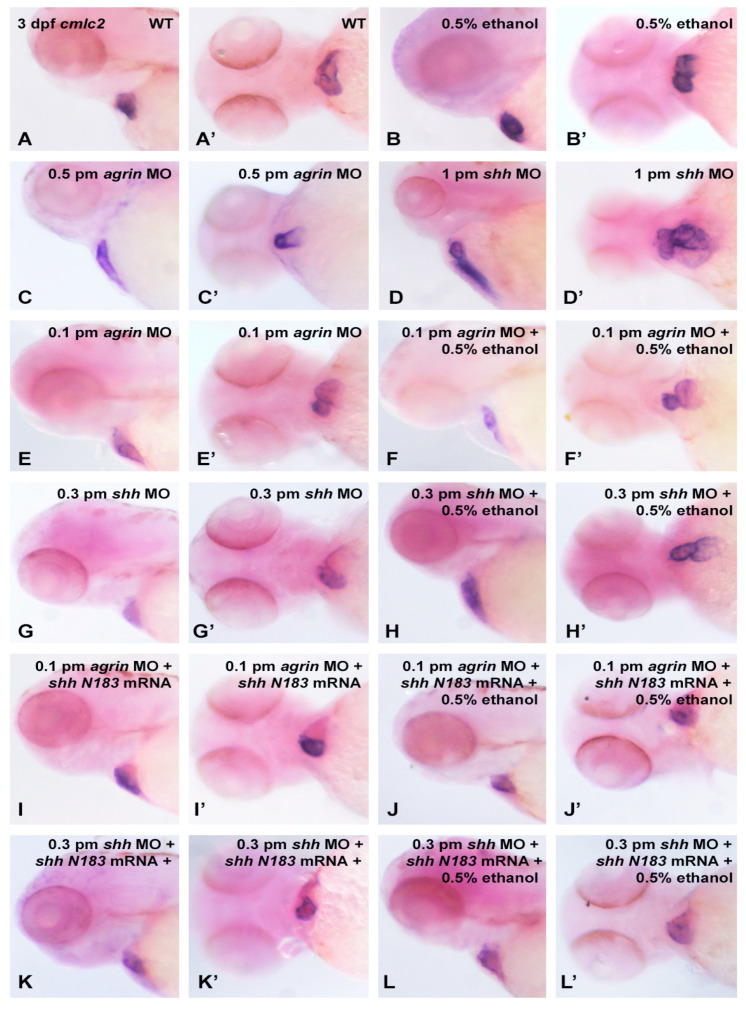
*Cmlc2* mRNA expression in 3 dpf zebrafish heart following MO and EtOH treatments. Ventral and lateral views of embryos are shown. (**A**,**A’**), wild-type; (**B**,**B’**), 0.5% EtOH exposure from 3–24 hpf; (**C**,**C’**), 0.5 pmol *agrin* MO injection of embryos; (**D**,**D’**), 1 pmol *shh* MO injection of embryos; (**E**,**E’**) 0.1 pmol *agrin* MO injection; (**F**,**F’**), 0.1 pmol *agrin* MO combined with 0.5% EtOH exposure; (**G**,**G’**), 0.3 pmol *shh* MO injection; (**H**,**H’**), 0.3 pmol *shh* MO combined with 0.5% EtOH; (**I**,**I’**), 0.1 pmol *agrin* MO and 25 pg *shhN183* mRNA injection of embryos; (**J**,**J’**), 25 pg *shhN183* mRNA injection rescues *cmlc2* expression pattern in embryos treated with 0.1 pmol *agrin* MO and 0.5% EtOH; (**K**,**K’**) 0.3 pmol *shh* MO and 25 pg *shhN183* mRNA injection of embryos; (**L**,**L’**), 25 pg *shhN183* mRNA injection rescues cmlc2 expression pattern in embryos treated with 0.3 pmol *shh* MO and 0.5% EtOH.

**Figure 9 cells-12-01327-f009:**
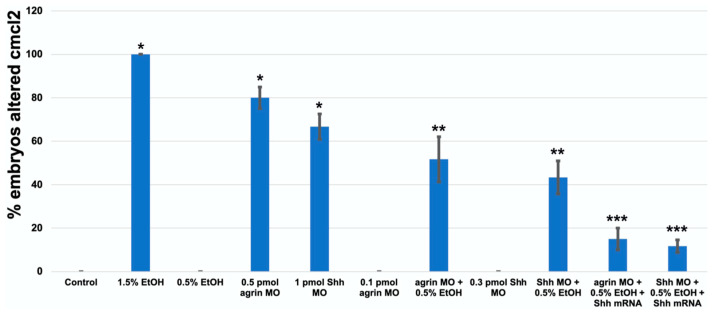
*Cmcl2* mRNA decreases, resulting from MO and ethanol exposures in 3 dpf embryos. * Significantly different from control *p* < 0.01; ** *p* < 0.025; *** Significantly different from MO + EtOH, *p* < 0.01.

## Data Availability

Data will be available upon request from the authors.

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
