# Peer review of "Embryonic Ethanol but Not Cannabinoid Exposure Affects Zebrafish Cardiac Development via Agrin and Sonic Hedgehog Interaction"

_cells, 2023, doi:10.3390/cells12091327_

Round 1

Reviewer 1 Report

The authors show that cannabinoid agonists do not affect heart development and that early ethanol exposure causes cardia bifida. They go on and show by sublethal ethanol concentrations and sublethal morpholino injections of shh and agrin have a synergistic effect on this phenotype. Overall it is a straight forward paper but with little mechanistic insight on how ethanol exposure might interact with shh and/or agrin.

The interaction between shh/agrin and ethanol has been described by the authors in the past (references 20,21) in the context of brain development Therefore, I find the synergy in early heart migration not really novel, especially at the level of morpholino - pharmacological treatments interaction   Having an antibody for agrin or shh (I know they are not easy to work) or showing at cellular or even transcriptional level (high resolution microscopy) how ethanol disrupts target proteins (for example fibronectin, S1P signalling or endoderm proteins have been involved in cardia bifida phenotypes). It is not clear if ectopic overexpression of shh in the one cell stage causes any developmental phenotype, besides rescuing cardia bifida   I selected major revisions just to give the editors the flexibility, depending also on other reviewer's comments to suggest new experiments or reject the manuscript.

Minor comment

Quantification of the data described for the combinations of treatments should be presented in a graph next to figure 6 and figure 7

Author Response

The authors' thank Reviewer 1 for this helpful review.

While our previous studies have described the role of agrin-EtOH interactions in brain development, the present study does provide new information, as we analyze cardiac marker genes and their dependence on agrin signaling.  We think these data are especially important when viewed in the context of the recent study that shows agrin injection in adult mouse heart, following myocardial infarction, leads to heart regeneration in an Erk-dependent mechanism.  Our previous studies in nervous tissue have shown that agrin markedly up-regulates Erk signaling.

We have not examined the genes indicated by Reviewer 1, but in our previous studies we have shown that Shh mRNA injection alone (the doses we employ) does not alter zebrafish morphology or alter expression of Mbx, Pax6 or GAD genes in zebrafish CNS.  In addition, the Shh mRNA overexpression in the presence of EtOH abolished the EtOH-mediated decreases in Pax6, GAD, Fgf19 and Atonal 1a gene expression in zebrafish CNS.  

Thus, the Shh mRNA rescue of altered cmcl2 gene expression pattern is consistent with our earlier CNS work.

We have now included graphical representation of the altered cmcl2 expression pattern in Figures 6 and 7, as suggested by Reviewer 1.  These are new figures, Figs 7 and 9.  The original Fig.7 is now Fig. 9.

Reviewer 2 Report

In this manuscript by Cole and coworkers, the authors study the possible influence of ethanol and cannabinoids on heart development in zebrafish, with specific attention to potential interactions with agrin and sonic hedgehog signaling.  In addition to the chemical exposure, the authors use anti-sense morphilino oligos against agrin and sonic Hh to examine synergy, and they use genetic rescue involving mRNA overexpression.  Sample size and statistical analysis seem appropriate.   Although the work does not break new ground, essentially a negative result with respect to cannabinoids type signaling on heart development, the study is solid and valuable and the manuscript is well written.  Would recommend publishing with only minor editing for typos (for example, see p5 line 178, replace grow with gross)  

Author Response

The authors' thank Reviewer 2 for the very positive review.  The typo has been corrected and the entire manuscript has been re-checked for any additional typos.

Round 2

Reviewer 1 Report

The authors edited several typos in the manuscript. Statistics of the quantifications for figures 6,7 now 7,9 are not described or depicted on the graphs.

Author Response

Thank you for your additional suggestion.  We have included error bars on Figures 7 and 9, as well as indicating statistical significance for high ethanol or MO alone, or low ethanol plus low MO.  The figures also show that the mRNA rescue is significantly different than the ethanol plus MO treatment.